# Synthesis of Doped/Hybrid Carbon Dots and Their Biomedical Application

**DOI:** 10.3390/nano12060898

**Published:** 2022-03-08

**Authors:** Vijay Bhooshan Kumar, Ze’ev Porat, Aharon Gedanken

**Affiliations:** 1Bar-Ilan Institute for Nanotechnology and Advanced Materials, Department of Chemistry, Bar-Ilan University, Ramat-Gan 5290002, Israel; 2Faculty of Life Sciences, Tel Aviv University, Tel Aviv 6997801, Israel; 3Division of Chemistry, Nuclear Research Center-Negev, Beer-Sheva 8419001, Israel; 4Unit of Environmental Engineering, Ben-Gurion University of the Negev, Beer-Sheva 8410501, Israel

**Keywords:** CDs, hybrid CDs, doped CDs, synthesis, characterization, biomedical, bioimaging, drug delivery, neuron tissue engineering

## Abstract

Carbon dots (CDs) are a novel type of carbon-based nanomaterial that has gained considerable attention for their unique optical properties, including tunable fluorescence, stability against photobleaching and photoblinking, and strong fluorescence, which is attributed to a large number of organic functional groups (amino groups, hydroxyl, ketonic, ester, and carboxyl groups, etc.). In addition, they also demonstrate high stability and electron mobility. This article reviews the topic of doped CDs with organic and inorganic atoms and molecules. Such doping leads to their functionalization to obtain desired physical and chemical properties for biomedical applications. We have mainly highlighted modification techniques, including doping, polymer capping, surface functionalization, nanocomposite and core-shell structures, which are aimed at their applications to the biomedical field, such as bioimaging, bio-sensor applications, neuron tissue engineering, drug delivery and cancer therapy. Finally, we discuss the key challenges to be addressed, the future directions of research, and the possibilities of a complete hybrid format of CD-based materials.

## 1. Introduction

CDs are part of nanoscale carbon materials, including carbon nanotubes, graphene, fullerene, nano-diamonds, and nanofibers. Over the past few years, carbon and carbon-based nanomaterials have attracted significant scientific attention for their physical, chemical, and biological properties [1]. Carbon-based nanomaterials are highly biocompatible, have low toxicity, and have impressive thermal and mechanical properties, as well as the ability to be functionalized with a wide variety of organic and inorganic molecules [2]. Carbon nanoscale materials with strong fluorescence are commonly referred to as CDs because of their unique properties. CDs are also characterized by their high stability, reduced harmful activity, water solubility, and derivatization ability [3,4]. These properties enable them to be used in a number of disciplines, as presented in Figure 1. Doped or hybrid CDs consist of graphitic carbon and functional heteroatoms (nonmetal) attached with a carbonized core [5,6,7,8,9,10]. These are a relatively new type of nanomaterials and are considered to be promising for various applications [11]. Nanomaterials called CDs are also refer to polymer dots, carbon quantum nanodots, and graphene or graphene oxide quantum dots.

In general, doped CDs are composed of sp^2^ hybridized conjugated carbon as the core and a shell of organic molecules with functional groups such as COOH, C=O, NH, OH, C-O-C, C-N, SH, or polymer aggregates [1,3]. Moreover, doped CDs are also considered the newest form of zero-dimensional carbon with less than 10 nm diameter and excellent fluorescent properties [12,13,14]. CDs have attracted considerable attention since their discovery in 2006 largely due to their unique fluorescence properties. It has found many applications in arenas of bioimaging [15,16], cell imaging [15], dye degradation [17], chemiluminescence [18,19], solar cells [20], photocatalyst [21], nanoelectronics devices [4], Combat COVID-19 [22], photodynamic therapy [23] and gene delivery [24]. Unlike pristine CDs, much less attention was dedicated to developing doped, hybrid or functionalized CDs either by metals, polymers or other atoms and molecules. Most of the papers dealing with this issue are related to the doped and hybrid CDs because of the enhanced fluorescence resulting from this doping [25,26,27,28,29,30].

CD-based materials are mainly synthesized by either top-down or bottom-up techniques [31], as presented schematically in Figure 2. According to the top-down approach, CDs are prepared by chemical and physical methods, including laser ablation/passivation, chemical oxidation, and electrochemical synthesis [32]. In the bottom-up approach, CDs are made from molecular precursors using laser ablation [33], pyrolysis [34], combustion [34], hydrothermal process [15], microwave [35,36], precipitation method [13] and sonication [37]. In contrast to bottom-up methods, top-down methods are not preferred because they require longer synthesis times, higher costs, and harsh environmental conditions [4].

Furthermore, the synthesis of doped CDs via top-down routes requires a separate step of doping and functionalization of the CD’s surface, whereas in the bottom-up methods, this is done simultaneously [38]. In addition to these two approaches, several others have been described, for example plasma treatment [39], cage-opening of fullerenes [40] and the solution chemistry synthesis [41]. These methods yield functionalized CDs in a wide range of sizes distribution and very hard to separate the well-dispersed CDs. There are also several other post-synthesis separation steps required, such as chromatography [42], dialysis [43], and gel electrophoresis [44].

Several non-metallic (nitrogen, sulfur, phosphorus, boron, fluorine) and metallic heteroatoms (Sn, Au, Ag, Zn, Ni, Mg, Mo, Gd, Cu, Ga, Fe, Co, Ca) were used as dopants. Some of them were found to affect the physicochemical characteristic of the doped CDs [4,45] The impregnation of metal or nonmetal ions may also change the electronic properties of doped CDs, designing the HOMO-LUMO energy band gap that determines the photo-optical properties of the doped CDs [6,46]. Doping with elements which are essential to the human body, such as calcium, iron, copper, manganese, copper and nickel made these doped CDs of special importance [10]. Doping with elements such as Au, Ag, Ga, Zn, and Fe are also of some importance for catalysis and biomedical applications. Compared to non-metal heteroatoms, metals are better electron donors, have a larger atomic radius and have a larger number of unoccupied orbitals [12,45]. Therefore, the charge density, as well as the charge transition between the graphene matrix and metallic ions, can be changed by doping the CDs with metallic ions [47]. Metal functionalized CDs have significantly more intense optical absorbance in the visible region than pristine CDs due to charge-transfer from the metal/metal-ion to the carbon or graphite [48]. As a result of the surface plasmonic resonance (SPR) of the metal nanoparticles, CDs also exhibit enhanced fluorescence properties [47]. Additionally, the presence of metal atoms in CDs results in an increase in their electrical conductivity as well as electron transfer and electron-acceptance capabilities, which facilitates electron transfer and increases their photocatalytic activity over pristine CDs [48,49]. Furthermore, CDs can act as an excellent reducing agent and reduce incorporated metal ions thus forming metal-doped CDs [50]. Conjugation of metal oxide nanomaterials with CDs or polymers improves their light-induced reactive oxygen species (ROS) production rate [5,51,52]. This enhancement cannot occur without the external stimuli of light. The light promotes up-conversion of the fluorescence characteristic, enhanced photo-induced electron transfer, and unique electron reservoir properties of the doped CDs. It also inhibits recombination of photo-induced electron-hole pairs [10,49,53]. All these effects occur due to the ROS generation. In addition, the increase in the concentration of ROSs enhances their photocatalytic antibacterial activities. Similarly, fluorescent CDs can be conjugated with magnetic nanoparticles to develop multimodal imaging platforms (optical and electromagnetic resonance). Ternary conjugates consisting of fluorescent CDs, NIR-responsive nanoparticles, and magnetic nanoparticles can be used in multimodal biomedical applications [10,49,53].

Many reviews have been reported on nonmetal-doped CDs and their applications [10,27,41,48,49,50,53] However, we found no review on the advancement of metal/nonmetal-doped/hybrid CDs, describing their physical or chemical properties and biomedical applications. The current review describes the recent progress in metal/nonmetal-doped/hybrid CD-based technology, considering their bottom-up synthetic procedures, the effect of doping or conjugation on the structure and the physical properties of the nanocomposites and their biomedical applications, which includes bioimaging, antimicrobial activity, drug delivery, cancer therapy, neuron tissue engineering and bio-sensor applications.

## 2. Synthesis of Hybrid and Doped CDs

Recently, several synthetic approaches for the synthesis of CDs and hybrid/doped CDs have been explored and reported. These approaches were aimed at improving the physical-chemical characteristics of the hybrid/doped CDs, such as increasing the fluorescence quantum yield and transparency and narrowing the band gap. These approaches are described herein. CDs can be synthesized by several methods, for example arc-discharge, laser ablation, sonochemical, microwave, hydrothermal and electrochemical methods and strong acid oxidation [4,54,55,56]. The first synthetic procedure was developed for the formation of pristine CDs by an arc discharge procedure [57,58,59]. It involved a polymerization reaction of small organic molecules followed by carbonization to form CDs. Generally, metal- and nonmetal-doped CDs have been classified into two groups: (i) top-down methods and (ii) bottom-up methods. Under the top-down synthesis method, larger carbon structures are broken down using chemical oxidation, discharge, the electrochemical oxidation of carbon, and ultrasonic methods [58,59]. However, this top-down approach has a number of disadvantages, such as harsh reaction conditions, expensive materials, and long reaction times [60,61,62]. As an alternative, a bottom-up approach involves converting carbon-based small molecules and structures into CDs of the desired size. Bottom-up methods include sonochemical, microwave, and hydrothermal thermal decomposition of organic molecules, pyrolysis of carbon-based materials, and solvothermal reaction synthesis [63]. In this report, we will discuss primarily the bottom-up approach to the synthesis of metal- and non-metal doped CDs using organic molecules and a variety of metal or non-metal precursors.

### 2.1. Synthesis of Metal-Doped CDs

Several bottom-up approaches have been extensively adopted for the creation of metal-doped CDs because of their low cost and better efficiency. CDs have been synthesized by solvothermal reactions of carbon-based materials and metallic salts to contain ions of silver [64], manganese [65], copper [49], magnesium [66,67], cobalt [68], zinc [69], gadolinium [28,70] and chromium [71]. As an example, aqueous GQDs (graphene quantum dots, which is the same as CDs) (0.1 mg/mL) were prepared by using an electrochemical cyclic voltammetry instrument [66]. Following this, a series of AgNO_3_ fresh aqueous solutions (0.1 mol/L) with measured volumes (5–60 mL) were added to CD (GQD) solutions (2 mL), respectively, and the above mixtures were stirred for 1 h to form a homogeneous solution of Ag-GQDs [64]. The resulting mixtures were then illuminated by UV light at the wavelength of 365 nm (18 W) from a distance of 20 mm for 3 h. When the photochemical reaction was completed, the color of the solutions transformed from yellow to brownish red (depending on the volume ratio of GQDs to Ag ions), indicating that Ag NPs had been formed [64]. The silver-doped CD (Ag-CD) nanoparticles on silica were engineered (Figure 3a) to form highly active surfaces enhanced by Raman scattering [64]. A number of researchers have also synthesized Zn-doped CDs for biomedical applications [69,72,73,74,75,76]. Dehydration, coordination, and carbonization are the steps required to synthesize Zn^2+^ doped CDs (Figure 3b) according to Cheng et al. [69]. In detail, Zn^2+^-doped CDs were prepared by dissolving citric acid, urea, and ZnCl_2_ in toluene. The resultant mixture was transferred to an autoclave hydrothermal reactor and heated at 200 °C for 12 h. It was then allowed to cool to room temperature. The brown solution obtained after the hydrothermal reaction contained the Zn ion-doped CDs. Evaporating the solvents led to the formation of solid samples, and the Zn^2+^ doped CDs showed excellent solubility in many organic solvents, as well as in water. In addition, we applied the as-synthesized Zn^2+^ doped CDs with intense bright yellow PL to a variety of applications. Additionally, Figure 3c shows different metal-doped CDs (M = Ga, Sn, Zn, Bi, Au, and Ag) and carbon dot-decorated substrates (Figure 3c) [4]. CDs were doped with low melting point metals by sonicating the molten metal overlayered with polyethylene glycol 400 (PEG-400) (PEG-400) [14,77,78]. The formation and doping of the CDs occurred simultaneously, and the effects of several experimental parameters, such as reaction temperature (4–150 °C), sonication duration (3 min–3 h) and sonication amplitude (40–70%), [37] were studied [79]. Doping the CDs with metallic Ga, In, Sn, Pb, Zn, Au or Ag was performed by ultrasound irradiation without the use of any catalyst or post-acid/base treatment [12]. Ga-doped CDs were also decorated with Ga nanoparticles and given the terminology Ga@CDs@Ga nanoparticles. Unlike the other reported synthetic routes for Metal@CDs the sonochemical synthesis. Sonications is the only direct technique for the preparation of Metal@CDs. Likewise, Sn-doped CDs [80] was synthesized by sonochemical reaction of polyethylene glycol-400 and metallic Sn as a precursor [78,79]. It yielded both Sn@CDs@Sn nanoparticles and Sn@CDs as depicted in Figure 3d. In a recent study, Xu et al., synthesized manganese-doped CDs (Mn-CDs) that demonstrated ultrahigh quantum yield of 54% for metal-doped CDs (Figure 3e) [76]. The Mn-CDs nanomaterials were prepared by a facile hydrothermal synthesis route using sodium citrate and manganese carbonate as the precursors and citric acid as an auxiliary agent. The materials displayed bright blue fluorescence and the capability to switch their luminescence on or off in the presence of compatible ions [76]. Taking advantage of the superior reversible fluorescence properties of the Mn-CD, Xu et al., developed a reusable universal flourescence particles for the nanomolar detection of a mercury ion in simulated polluted water samples [76]. Moreover, Duan et al., described a facile solid-phase synthesis method for the Cu-doped CDs nanomaterials using citric acid as the carbon source and Cu(NO_3_)_2_·3H_2_O as the dopant materials, respectively [49]. These Cu-CDs show superior peroxidase-like activity to horseradish peroxidase and were stable under a wide range of pH and temperatures [49].

### 2.2. Synthesis of Nonmetal-Doped CDs

The Doping of nonmetals in CDS can affect the overall electrical distribution and the relevant levels of electronic energy in the nonmetal-doped CDs [81]. CDs can be tuned to exhibit their desired fluorescent and other properties by doping foreign atoms [81]. In order to improve the properties of CDs for further applications, various nonmetals such as boron [82], nitrogen [83], phosphorous [84], sulfur [85], silicon [86], and halogens (F, Cl) [87] have been doped into the CDs, using either the “top-down” or “bottom-up” approaches. B, N, and P- doped CDs were recently synthesized by Kumar et al., using hydrothermal and sonochemical methods [45]. They synthesized N-doped carbon quantum dots (NCDs) from bovine serum albumin protein in water via a hydrothermal chemical reaction [15]. The involved the steps of fragmentation, aromatization, polymerization, and carbonization (The synthesis procedure are depicted in Figure 4a) [88]. During the hydrothermal reaction (>175 °C) carbonization of the BSA polymer occurs leading to the formation of a final NCDs material with a complicated surface structure [88]. The same hydrothermal method was used to synthesize boron-doped and phosphorus-doped CDs [45]. A temperature-dependent mechanism for the synthesis of N-doped CDs has been proposed. These NCDs were used for doping/functionalization of hydroxyapatite (NCDs-HA) nanomaterials in water by stirring of hydroxyapatite (HA) with NCDs at room temperature [89]. Figure 4b presents a schematic illustration of doping NCDs with the HA particles. NCDs were used as the stabilizer in three different compositions of HA particles accommodate 1% to 5% NCDs [89]. Adding P and N did not improve the fluorescence properties of N/P-co-doped CDs, although their aqueous dispersibility increases significantly [90]. The purpose of S-doping in CDs is typically to enhance metal ion (Fe^3+^ ions, other heavy metal ion) detection sensitivity by 0–500 µM [91]. However, if the S atom is fixed to the ring of polythiophene, strong red-emitting light can be observed when the excitation wavelength is 543 nm [92]. Further, the B/ N/S-co-doped CDs demonstrate high-efficiency red emission at wavelength near 600 nm [92]. Even though a variety of nonmetals, such as N, P, S, and B, have been reported, there is still a strong demand for the development of highly efficient CDs in term of real application for our society [92]. Therefore, further exploration of other nonmetal doping systems is highly desirable. For example, although fluorine is not present in biological systems, the incorporation of F-containing grafts can increase the therapeutic efficacy of many drugs, improve the chemical stability of proteins, and enhance phase separation in both polar and nonpolar conditions [93]. Thus, by modifying CDs with the F, it may be possible to significantly tune their fluorescence properties including wavelengths and behavior in biological systems or biomedical applications [93]. Zuo et al., studied the F-doping strategy to dramatically lengthen the emission wavelength of CDs [94]. Using 1,2-diamino-4,5-difluorobenzene as the fluorine source and tartaric acid to improve aqueous solubility, they synthesized a kind of F-doped CD (F-CD) in a one-pot solvothermal process [94]. When compared with undoped CDs, the emission wavelength of F-CDs is redshifted by 50 nm [63,94,95,96]. Using an excitation wavelength of 480 nm, the F-CD emits intense yellow fluorescence, while the emitted red light is found upon excitation at 540 nm [94]. Additionally, the F-CDs provide highly sensitive detection of intracellular Ag+ as well as high imaging sensitivity of red blood cells in various cell systems [94]. The hydrothermal method was by Kalaiyarasan to synthesize phosphorus-doped CQDs (PCQDs) from phosphoric acid and Trisodium citrate (TSC) (Figure 4d) [97]. They investigated the effect of the reaction temperature, durationand precursor concentrations on the formation of PCQDs and the doping effect of phosphorus in CDs. and found that fluorometric probes for iron detection. Moreover, the P-CDs with high fluorescence QY have been examined for changes in the fluorescence intensities induced by the addition of Fe^3+^. These changes are a consequence of the formation of a stable fluorescent inactive complex (P-CDs-Fe^3+^), which may be useful for biomedical applications (Such as live cell imaging, Fe^3+^ detection in blood, urine) [97]. In addition, a highly sensitive fluorescent probe based on sulfur-doped carbon dots (S-CDs) was synthesized by Kamali et al., using a microwave irradiation method (Figure 4e) [36]. Using this probe, a strong blue emission was observed with 36% QY [36]. This class of nonmetal-doped CDs (B, N, P, S, F, etc) or co-doping two/three nonmetals was utilized as an effective fluorescent materials for live cell imaging owing to their excellent QY, good water dispersibility, tunable strong fluorescence, better biocompatibility and low cytotoxicity.

### 2.3. Nanohybrids of CDs with Metals and Metal Oxides

CDs-metal nanohybrids are prepared mainly by mixing CDs with dispersions of metallic nanoparticles followed by sonochemical reaction. Kumar et al., synthesized TiO_2_/Sn@CDs hybrid nanomaterials by adding TiO_2_ nanoparticles in the powder form into an aqueous solution of Sn@CDs and sonicating for 20 min, followed by drying at 80 °C for 12 h in a vacuum chamber [17]. Recently, Li et al., prepared Cu_2_O/CDs hybrid nanomaterials using ultrasonic reaction [98]. Kumar et al., also developed hybrid nanostructured CDs decorated with Fe_3_O_4_ for neuronal growth enhancement [13]. These Fe_3_O_4_ functionalized CDs nanospheres were dispersed in an aqueous solution by ultrasonication for 30 min for further study the physical, chemical and biological properties. Tang et al., prepared CDs-BiVO_4_ nanosphere by refluxing BiVO_4_ powder in an aqueous solution of synthesized CDs at 90 °C for three hours. Jiang et al., prepared gold-doped CDs (Au@CDs) or gold decorated CDs nanomaterials by using anisotropic gold nanomaterials via microwave assisted method [99]. Tiny gold clusters were found scattered within the skeletons of CDs or on the surfaces of CDs. Au@CDs effectively combine the properties of CDs and gold nanoclusters (AuNCs). In order to obtain the nanoprobes, antibodies were attached to the surface of Au@CDs for further biomedical applications, especially intracellular imaging of cancer-derived exosomes) [99,100].

## 3. Physical and Chemical Properties of Doped and Hybrid CDs

CDs are particles of 2–10 nm in size, with a sp^2^ graphitic carbon core and functionalized with polar oxygenators groups. In the UV-Vis absorption spectrum, they show two types of absorbance peaks: one at ~240–260 nm and the other at 290–360 nm. These are assigned to the π–π* transition of sp^2^ core carbons and the n–π* transition carbonyl organic functional groups of CDs, respectively. Kumar et al., reported the fluorescence properties of synthesized N-doped CDs prepared from protein (bovine serum albumin). The excitation wavelengths (λex) were from 330 nm to 470 nm and the emission revealed excitation-dependent properties. The fluorescence excitation-dependent emission of the N-doped CDs showed narrow bands, with the maximum emission intensity at 462 nm following excitation at 390 nm. In other words, the particle size distribution is relatively narrow (Figure 5A) [88]. FTIR analyses showed that the CDs contains mainly hydroxyl, carboxyl, carboxyl and epoxy oxygenator functional groups on its surface. Generally, the FT-IR spectrum of doped CDs is slightly different from that of pristine CDs: the stretching vibrations can be found at 3492, 2935, 1730, 1625, 1422, 1264, and 918 cm^−1^ due to the surface presence of OH groups, C-H groups, C=O, C=C, C-O-C, and epoxy groups, respectively [101]. The XPS analysis confirmed that carbon, nitrogen, and oxygen are present in the N-doped CDs, as the peaks at 384 eV and 532 eV correspond to the binding energies of these elements (Figure 5C) [88].

A high-resolution XPS spectrum of C 1 s and O 1 s revealed that the presence of several functional organic groups, such as C-C, C-N, C=C, C=O, C-O, and C-OH, are present on the surface of N-doped CDs. In addition, nitrogen-containing groups were also detected. This phenomenon may be due to the presence of nitrogen-containing precursors or to the presence of doping or functionalizing agents in CDs [88]. Kumar et al., analyzed the graphitic nature of carbon in N-doped CDs by Raman spectroscopy. The Raman spectrum of N-doped CDs (Figure 5B) shows two major bands: a D-band at approximately 1346 cm^−1^, which is attributed to the presence of SP^2^ carbon defects, and a G-band at approximately 1566 cm^−1^, which is attributed to the stretching vibration of C-C graphite. The results of these experiments indicate that N-doped CDs contain graphitic carbon. According to previous studies, TEM images revealed spherical N-doped CDs particles with relatively narrow size distributions (Figure 5D,E) and an average size of about 5 nm [5,71,102]. Based on DLS analysis, N-doped CDs exhibited a similar particle size (3−8 nm), as shown by TEM analysis. CDs and N-doped CDs were found to be chemically and physically stable at pH values of 3.5, 5.0, 7.0, and 10.0 [16]. The size and morphology of the various metal-doped CDs (Ga, Sn, Ag, Zn and Au) were studied by high-resolution TEM [12], showing spherical shaped in the size in the range of 3–7 nm. EDS and XPS analyses provided the elemental composition of the CDs; only C and O with traces of the doped metal/nonmetal were found, as expected [12]. The fluorescence was affected by metal doping, whereas the lattice parameters were not changed [12].

Figure 6 shows the fluorescence pattern of pristine CDs and several metal-doped CDs at different excitation wavelengths. Figure 6a shows the typical high-fluorescence spectra obtained from the pristine CDs. The Metal-based nanoparticles alone showed no fluorescence, whereas the fluorescence of all the other examined metal-doped CDs (Sn, Zn, Ag and Au) was less intense than for the pristine CDs (Figure 6b–f). The reduction of fluorescence intensity of N-doped CDs is thus an unintended indication for the incorporating of the metals in the synthesized CDs [12]. However, nonmetal (B, N, P.) doped CDs exhibited higher florescence than pristine CDs due to lower atomic weight element [45].

## 4. Applications of Doped CDs

CDs and CD-based nanomaterials have a unique combination of physical, chemical and biological properties, which make them suitable for a broad range of applications. The prospects of using C-based quantum dots (QDs) offer the advantage of the absence of pristine heavy metal atoms (typically toxic metals), which limits the use of QDs in biomedicine. The low doping of metal ions or atomic metal in CDs can make nontoxic hybrid or doped CDs that can be suitable for performing in vivo and in vitro bioimaging and drug delivery experiments. There have been many reports on the potential biological applications of doped CDs as antiviral, antimicrobial, antiparasitic, and antibacterial agents, and even against mosquitoes. Most doped CD- and hybrid CD-based strategies have been designed and tested solely for laboratory experiments. In the following sections, we describe a variety of possible applications of doped/hybrid CDs in terms of their technological versatility.

### 4.1. Doped CD-Based Sensor Applications in Biology

CDs and metal/nonmetal-doped CDs are considered potential optical sensors due to their excellent photostability and numerous binding sites. The organic molecules and other compounds can be detected in a variety of ways, including photo-induced electron transfer (PET), charge transfer, fluorescence resonance energy transfer (FRET), and the inner filter effect of metal-doped CDs.

#### 4.1.1. Optical Sensor for Biological Applications

Waste watermelon peels were pyrolyzed at low temperatures and filtered to produce large-scale N-doped CDs [103]. Such CDs have strong blue luminescence, are water-soluble and stable in solutions at a wide pH range (Figure 7a). They can be used to stain HeLa cells for imaging. (Figure 7a) [103]. The pyrolysis of peanut shells resulted in N-doped CDs with excellent stability, better photobleaching resistance, and superior tolerance to pH variation and different ionic strength [104]. These CDs exhibited low inherent cytotoxicity and a quantum yield of 10% of fluorescent (Figure 7b). [104].

#### 4.1.2. Fluorescent Metal Doped CDs for Biosensor Applications

Ge-doped CDs (Ge-CDs) were recently produced using organic citric acid compound and ^132^Ge as a precursor. The Ge-CDs had a QY of about 9%, while similarly produced undoped CDs showed a QY of about 6% [105]. A simple carbonization procedure was used to produce Ge-CDs nanomaterials in 15 min with minimal cell toxicity, high intracellular drug delivery efficiency, superior biocompatible and high stability [105]. Yan et al., shows how the addition of Hg^2+^ causes the formation of Ge-CDs. Moreover, the prepared Ge-CDs show better specific binding ability to Hg^2+^, which may be useful for the detection of this toxic element even in extremely complicated media, such as honeysuckle condensation samples. In particular, the synthesized Ge-CDs can be applied for bioimaging of cells and monitoring the cellular activities in Hep-2 living cells [105]. Consequently, Ge-CDs may offer great potential for real-time monitoring of Hg^2+^ ion in the living cells as a better-performance platform. The quenching of the fluorescence intensity is due to the interactions between Hg^2+^and the-OH molecules and -COOH functional groups on the surface of synthesized Ge-CDs. Adenosine disodium triphosphate (ATP) compounds and LaCl_3_ were used to fabricate blue fluorescent La-CDs by solution chemistry. Under high-salt conditions, the La-CDs demonstrated good stability [106,107,108]. Hg^2+^ ions were chelated with –COOH, -PO_4_^3−^ and –NH_2_ groups after addition to La-CDs suspension, resulting in a charge transfer mechanism that resulted in the quenching of fluorescence intensity. It was found that the synthesized La-CDs were highly selective and sensitive towards Hg^2+^, with a detection limit of 0.1 µM in living cells or tissues [106,107,108]. Additionally, Zhang et al., demonstrate that La-doped CDs have strong fluorescence and optical monitoring codes providing superior accuracy both in vitro and *in vivo*, and enhance blood compatibility [109]. Wang et al., have recently developed an advanced hydrothermal autoclave reaction to prepare lanthanum nanoparticles with N- or P- doped carbon dots (La-N/P-CDs) to produce N/P-CDs with antibacterial properties [110].

A microwave-assisted hydrothermal reaction in autoclave was used to fabricate Gd-doped CDs (Gd-CDs) with super-high quantum yields [28]. Purification of these Gd-CDs by centrifugation is relatively easy, because of their small diameters (4–8 nm). The researchers developed a “turn-off-on” fluorescent biosensor that detects glucose very accurately using Gd-CDs as probes [28]. Their fluorescence was reduced by the carbon microparticles-glucose oxidase complex molecules (CMP@GOx). Hu et al., have illustrated the assembling interaction between the CMPs@GOx molecules and synthesized Gd-CDs which causes the quenching [28]. Glucose, however, can enter into the space between the synthesized Gd-CDs NPs and CMP@GOx molecules, leading to the release the Gd-CDs nanoparticles, thereby disrupting the FRET development and allowing the Gd-CDs to fluoresce again [28]. In addition to serving as a binding site for analytes, the metals in the metal-doped CDs are used as coordinate binding sites [8]. Lu et al., studied the fluorescence of chromium (III)-doped CDs, which have low cytotoxicity [71]. With a graphene-analogous structure, these Cr-CDs emit greenish blue fluorescence with excitation and emission maxima at 350/466 nm, having 20% fluorescence quantum yield with excitation-independent emission properties [71]. Addition of para-nitro phenol (p-NP) resulted in fluorescence quenching of the greenish blue emission of the Cr-CDs, whereas the lifetime of the fluorescence remained unchanged. Since the absorption spectrum of p-NP and the excitation spectrum of Cr-CDs overlap, the reduction of flourescence mechanism was assumed to be inert and the inner filter effect [102]. In a recent study [9], the solvothermal carbonization of folic acid and CuCl_2_ yielded copper-doped nitrogen-carbon dots (Cu-NCDs). These exhibited unique physico-chemical, optical, electrical, and biological properties due to their Cu doping and surface modification of Cu-NCDs (Figure 7c). The Cu-doped CDs showed double emission peaks at 410 nm and 470 nm, with the original peak attributed to structural and surface changes and the surface passivation with different organic molecules. They showed also good quenching responses to ascorbic acid (AA) in the concentration range of 0.02–40 µM, and limit of detection (LOD) of 17.8 nM (Figure 7c) [9]. The Cu-NCDs were also studied in the bioimaging of HepG2 cells. Based on these results, doped CDs are potentially useful for bio-sensing and multifunctional applications.

#### 4.1.3. Colorimetric Biosensors Applications

Colorimetric sensors can detect toxic pollutants visually [52]. Mn-doped CDs (Mn-CDs) showed oxidase-like chemical reactivity and were accomplished of oxidizing tetramethylbenzidine (TMB) into oxTMB [111]. As a result, the colorless solution turned blue, giving an absorption band at 652 nm [111]. As a result of a redox reaction, addition of ascorbic acid (AA) led to disappearance of the blue color. The synthesized Mn-CDs exhibited excellent sensitivity to AA with detection limit of 9 nM [111]. Similarly, a colorimetric-based sensor of Mo-CDs particles was developed that exhibited peroxidase activity for the detection of cholesterol [112]. As shown in Figure 7d, Mo-CDs particles easily oxidized TMB to oxTMB when exposed to H_2_O_2_ and the colorless solution turned blue. The cholesterol oxidase (Chox)-triggered oxidation of cholesterol could be detected using a Mo-CD-based cascade colorimetric biosensor. An enzyme-triggered reaction was monitored by a Mo-CDs with high catalytic sensitivity to H_2_O_2_ [112]. Using the Mo-CDs-based biosensors for cholesterol detection, they have demonstrated excellent sensitivity and selectivity (LOD of 7 µM). This is due to the Mo doping, which allows the electrons to be transferred more easily and thus enhance the CDs’ catalytic activity. Chen et al., synthesized CDs@5′-adenosine monophosphate with Eu nanocomposite (AMP/Eu), which showed two main emission peaks at 430 nm and 615 nm following excitation at 310 nm [108]. With the addition of oxytetracycline (OTC), the fluorescence of Eu^3+^ at 615 nm was enhanced, while the fluorescence of CDs (at 430 nm) was quenched since the absorption spectrum of OTC overlapped both the excitation and emission spectrum of CDs [108].

**Figure 7 nanomaterials-12-00898-f007:**
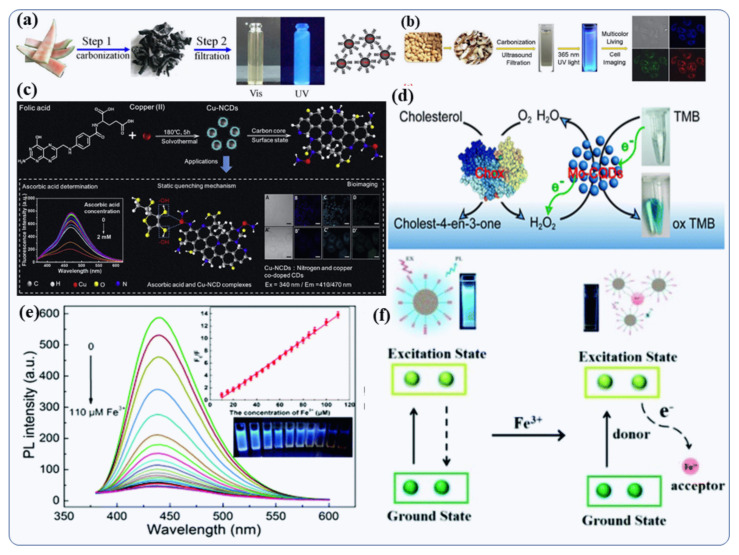
Preparation of fluorescence doped CDs by pyrolysis reaction method. (**a**) Development of water-soluble N-doped CDs from watermelon peel [103]. (**b**) An overview of the preparation and application of N-doped CDs derived from peanut shells [104]. (**c**) Cu-CDs and their applications for the detection of Hg^2+^, glucose, p-NP, and AA, respectively [9]. (**d**) Colorimetric detection of cholesterol by Chox and Mo-CDs particle [112]. (**e**) Ultraviolet-visible absorption spectra of Fe-CDs with different concentrations of Fe^3+^ ion [113]. (**f**) FeCDs particles for detection of Fe^3+^ using dual-mode fluorometric and colorimetric sensing [113]. Reproduced with copyright permission.

Fe-CDs were prepared by Zhao et al., from ammonium iron (III) citrate and urea as precursors. via hydrothermal reaction at 180 °C for six hours in stainless steel autoclave [114,115,116,117,118,119]. The products were then cooled to room temperature and neutralized to pH 7.0. The Fe-CDs proved to be promising dual-mode fluorometric and colorimetric nanosensors for Fe^3+^ detection [113]. Figure 7e,f show that the blue photoluminescence of Fe-CDs was quenched upon addition of Fe^3+^ ions and the absorption band at 335 nm was increased [113]. The authors suggest that the actual coordination exchanges between Fe^3+^ and the organic functional groups on the surface of the Fe-CDs contribute to the formation of the conjugate structure. According to He et al., Fe-CDs-Au Nano crystals (NCs) nanoprobes were prepared for ratiometric fluorescence detection of Zn^2+^ at two emission wavelengths (445 nm and 650 nm) [113]. It was found that the red fluorescence of the Au nanocrystals increased with increasing Zn^2+^ concentration, whereas the blue fluorescence of the Fe-CDs nanoparticles was independent [113]. This has been ascribed to the AIE enhancement of the Au nanocrystals. Adding Zn^2+^ to the solution mixture caused the fluorescent color to change from blue to pink.

### 4.2. Biochemical or Biomolecular Sensing

Doped CDs can be used in biological sensing due to their excellent electronic and electrochemical properties. Doped or hybrid CDs are utilized for photoluminescence, electrochemiluminescence and electrochemical sensors [114]. Benefitting from the turn-off of the photoluminescence properties of CDs, biological ions, biomolecules (sugar, antibodies, other sugars, amino acid) and proteins can be detected [115,116,117]. Cu^2+^ and Fe^3+^ ions play a crucial role in detection of diseases [118,119]. Hence, CDs can be used for detecting such ions which are co-factors for specific diseases. To detect a particular gene classification, even enzymatic reactions cannot be used. Since DNA has a highly specific nucleotide sequence, the principle of complementation is the only viable solution [51]. A combination of novel CDs and DNA probe for biosensor activity has been received wide-ranging interest in recent years due to their better functionalized properties of CDs and their superior affinity towards DNA, either by assembling or electrostatic charges interaction. Using the principle of DNA complementation, CDs coated with DNA were probed and placed on an Au nanoelectrode to create a DNA biosensor [120]. Nucleic acid biosensors often use the “on-off-on” strategy, just like other types of biosensors. In the case of DNA hybridization between two DNAs (single-stranded), the binding force is considerable greater than that of a single strand DNA with a quencher [120]. This is caused by the unique matching structures of DNA strands. Since the DNA strand has a strong tendency to complicate with the quencher, even the already-quenched CDs PL can be recovered in the occurrence of the DNA target strand (Figure 8a) [51]. In a hybrid CD-based optical biosensor, the degree of fluorescence reduction is used to detect target analytes. As an example, Shen et al. [115] designed a glucose biosensor based on phenyl-boronic acid-derived CDs particles. The surface boric acid of the CD interacts with glucose, allowing FRET energy transfer between the CD and glucose, effectively quenching the CD’s fluorescence (Figure 8b) [51]. As a result of the degree of fluorescence quenching, the body’s glucose levels are assessed, achieving a 10–250 times higher sensitivity than the previous boric acid fluorescent nano-sensing exposure system [51].

Recently, Liu et al., described the facile synthesis of N-doped CDs (5 nm) with the assistance of a new passivation mediator poly (acrylate sodium) for sensing applications [121]. Experimental results of intracellular have demonstrated that a folic acid (FA)-based CDs probe can precisely differentiate folate receptor (FR)-positive cancerous cells from normal cells in cell mixtures as shown in Figure 8c [121]. Furthermore, Jia et al., studied the formation of magneto-fluorescent Mn-CDs from manganese (II) phthalocyanine for cancer cell biosensors. Significantly, the Mn-CDs not only effectively produce reactive oxygen species (ROS) but also highly catalyze the generation of oxygen. Thus, Mn-CDs is potentially a good biosensing material for tumor hypoxia and improved photodynamic therapy (PDT) [122].

Wang et al., have reported that in the laboratory, the unique binding of mannose units to E. coli enabled quantitative detection at levels as low as 450 colony forming units (CFU)/mL [123]. This experiment enabled the application of mannose-based CDs to actual bacterial analyses. Later, Chen et al., studied the sensing of the E. coli bacteria O157: H7 using water-soluble N-doped CDs as a probe [123]. They observed that the E. coli bacteria caused an improved effect on the photoluminescence emission of N-doped CDs particles with a linear range of 108 CFU/mL for study the antimicrobial properties [123]. Modifying CDs with receptors is necessary and has been widely utilized by other researchers to detect bacteria selectively and sensitively. Nandi et al., developed a simple, analytical platform for the detection and imaging of bacteria that is based on the attachment of amphiphilic doped CDs to bacterial cells (Figure 8d,e) [124]. The functionalized CDs containing hydrocarbon chains were readily bound to bacteria following a short incubation and allowed for the detection of bacteria using fluorescence spectroscopy and microscopy [124]. It could be possible to distinguish bacteria even in a mixture of populations by the intensity and spectral position of CD fluorescence depending on the microbial species.

**Figure 8 nanomaterials-12-00898-f008:**
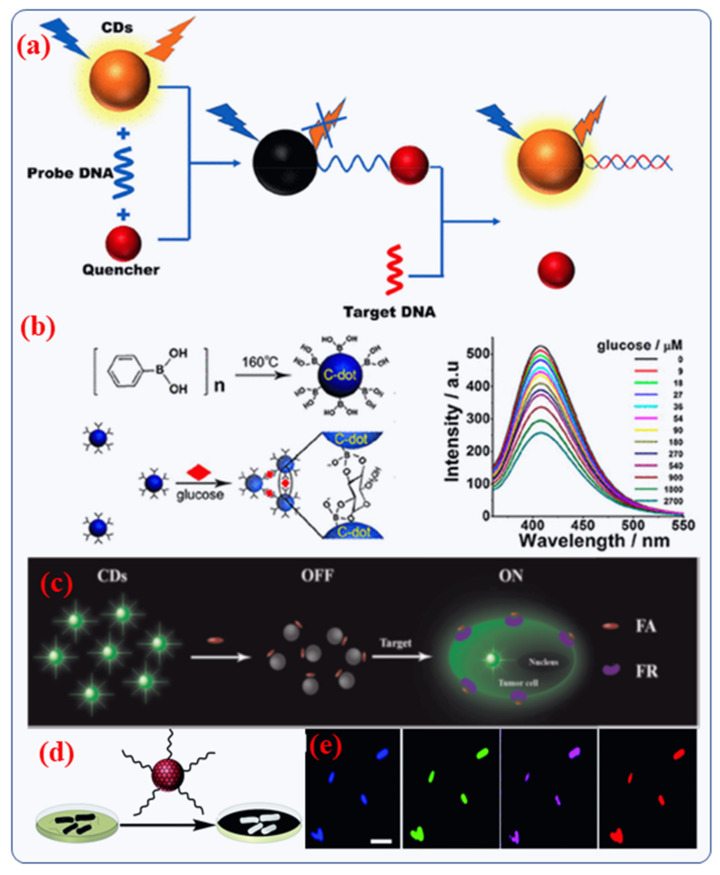
(**a**) Schematic presentation of the sensing approach of a functionalized CD biosensor that depends on DNA interaction. Copyright with permission from Ref. [51], from American Chemical Soceity (**b**) Recognition of glucose by a functionalized CD biosensor via FRET [115]. Copyright with permission from Ref. [115], from American Chemical Soceity. (**c**) Schematic presentation of the designed fluorescence-based biosensor for turn-on probe for overexpressed cancer cells. Copyright with permission from Ref. [121], from Elsevier (**d**,**e**) The detection method utilizes fluorescent CDs that are used to label bacteria and multi-color fluorescence microscopy images of E. Coli. Copyright with permission from Ref. [124], from Royal Society of Chemistry.

### 4.3. Other Nano Sensing Applications

The literature describes nanosensors based on hybrid or doped CDs that were designed by two different approaches: 1. sensors consisting of “hybrid/doped” CDs via specific target molecules and 2. functional CD nanocomposites, which can bind the target chemicals or other nanomaterials. The use of doped CDs as “raw” sensors is reported in the literature. Differently-prepared doped CDs can be applied to detect various metal cations such as Hg^2+^ ion, Cu^2+^ ion and Fe^3+^ ions, as well as small or macro-molecules in solution, even in the biomedical context [25,125]. Sensing occurs via the emission of quenching/enhancing due to the detachment of the nanoparticle from the CD surface, induced by the existence of the analyte.

### 4.4. Bioimaging of Doped CDs

Bioimaging uses organic and inorganic fluorophores, which induce weak fluorescence and thus require binding with antibody molecules for cellular uptake. [126]. Metal-doped CDs, due to their exceptional optical properties, tissue compatibility and efficient cellular uptake, are gaining popularity as a substitute to organic/inorganic fluorophores [126]. The biological applications of metal-doped CDs show promising results when formulated by the sonochemical method. This is due to their high biocompatibility whereas cytotoxicity can be induced during chemical preparation via the use of toxic chemicals and adverse conditions [78]. Metal-doped CDs were used as fluorescent tags for cell imaging by Sun et al., [127]. Furthermore, metals-doped CDs (Ga, Ag, Au, Sn, and Zn) have been widely used to image intracellular diverse cells, such as HeLa cells, HepG2 cells line, and neuronal cells line such as PC12 and SH-SY5Y (Figure 9) [12]. The optical properties [128] of doped CDs were also used for real-time molecules stalking in living cells, due to their superior photostability, biocompatibility and bright fluorescence [128]. Zheng et al., used the CDs to conjugate with protein for real-time monitoring [129]. When imaging fluorescent probes in vivo, long wavelengths are crucial, and they must be transparent to biological tissues in comparison to cell imaging and cellular uptake. Due to their multicolor fluorescence properties, hybrid or doped CDs are a good choice for in vivo imaging. Huang et al., described in vivo imaging with N-doped CDs injected to mice [130].

### 4.5. Doped CDs for Live Cell Imaging of Cell/Nucleus/Tissue and Living Organism

Kumar et al., presented a single-step synthesis strategy for N-doped CDs (N@CDs) with high quantum yields (44%), better photostability, high colloidal stability and excellent functionalization efficiency [15] In addition to their cost effective, the N@CDs were non-toxic and long-lasting when used for live-cell imaging of human U2OS cells (Figure 10a). Due to their high biological stability, high tissue compatibility and homogenous dispersal, the N@CDs suspension was used for high-quality cell imaging and to model the biological effects on the nuclear level [15]. Gd-doped CDs were found to be useful as theranostic agents, as demonstrated by Du et al. [70]. An MTT (3-(4,5-dimethylthiazol-2-yl)-2,5-diphenyltetrazolium bromide or MTT) assay was used to test Gd-doped CDs for their nontoxicity to human adenocarcinoma lung cancer cells and vascular smooth muscle cells (VSMCs) [8,9.131]. Furthermore, bright-field images showed no morphological modification after Gd-doped CDs treatment, thus suggesting good biocompatibility [70]. Using the fluorescence property of the Gd-doped CDs, Du et al., evaluated their uptake into living cells [70]. The labeled cells (HepG2 cells and VSMCs) exhibited a green fluorescence produced by Gd-doped CDs dispersed in the cytosol (Figure 10b) [70]. They used H&E cell line and anti-Ki67 Z (cancer cell marker), to confirm the diagnostic accuracy. Figure 10c depicts the H&E and immunohistochemistry images that confirmed the accurate detection of Gd-CDs enhanced MRI analysis showing pathological tissue changes along with hepatocellular cancer cell in the subcutaneous (skin) tumor samples and liver. Studies have shown that Gd-CDs are extremely sensitive for MRI imagining of tumors [70]. A number of studies showed that long circulation times are beneficial to passive targeting of solid tumors through leaky vasculature. The understanding of the Gd-CDs’ biodistribution behavior into the major tissues would be crucial for their further in-vivo medical applications [70]. An ICP-MS analytical tool was used to examine the metabolism and distribution of Gd-doped CDs in vivo (Figure 10d). Particles have a significant uptake in the spleen and heart for 24 h after injection for the bio-imaging. The extremely small size inorganic nanoparticles (less than 5 nm) allowed a percentage of the particles to escape to recognition by the reticuloendothelial system and collect in different organs such as the liver, lung, and kidney. There may be differences in the distribution of Gd-CDs due to the abundant blood supply in the heart and the presence of the reticuloendothelial system in the spleen. Moreover, the Gd-CDs did not cause apparent adverse effects, such as tissue injury, swelling or lesions when compared to the control group after 7 days post-injection (Figure 10e) [70]. Furthermore, N-doped CDs are very stable, long-lasting and have no detectable toxicity [88,89]. Using N-doped CDs, Kumar et al., validated this concept on zebrafish (ZF) embryos, which were incubated in aqueous solution containing N-doped CDs, as shown in Figure 10f [88]. As a result of their ultrafine size, N-doped CD particles readily enter into ZF embryos through the germ ring around the yolk sac and chorion of embryo during incubation. The ZF embryo chorion pores’ size is 170 nm, which is much larger than N-doped CDs (~5 nm) and allows the penetration of CDs. At 24 hpf (hours post fertilization), ZF embryos exposed to 6 mg mL^−1^ N-doped CDs solution exhibited strong fluorescence in the yolk sac, trunk and tail region, indicating the tissue-dependent affinity of these CDs [88]. The other remaining portions of the embryo (zebra fish) exhibited weak N-doped CD fluorescence emission at 24 hpf (Figure 10f) [88].

In addition, at 48 hpf, when the embryos crosshatched into larvae, N@CD fluorescence was observed primarily in the yolk sac of ZF larvae (Figure 10g). Furthermore, fluorescent signals were observed in the ZF head (mostly in the eyeball) and on the longitudinal melanophore strip at the rear part of the ZF body (Figure 10h) [88]. These results indicate that N@CDs do not have a toxic effect on embryo development. Thus, bioimaging of zebrafish shows that N-doped CDs have potential bioimaging applications in diverse aspects of lipoprotein in a zebrafish yolk lipid metabolism and transport model. N-doped CDs can simply cross the blood ocular barrier on the specific fluorescence of the ZF larva eyes, where the lens can easily be seen inside the eyeball (Figure 10g,i) [88]. However, the absence of fluorescence in the larval head of the ZF indicates that the N-doped CDs were not able to cross the blood-brain barrier. This is similar to other vertebrates in terms of ocular development. Therefore, N-doped CDs could be useful in eye-related disease diagnostic and live cell imaging. Our hypothesis is that the selective accumulation of N-doped CDs is due to their interactions with the melanin present inside the pigmented cells (melanocytes) in the eyes, as well as the melanophore strips found on the tail and trunk of the ZF larvae [88]. Our hypothesis was based on the observation that the binding of N-doped CDs to melanin is due to the strong interaction between the organic functional groups of melanin and N-doped CDs shown in Figure 10j.

### 4.6. Metal-Doped CDs for Neuron Tissue Engineering

The regeneration of neuroblastoma cells or neurons is one of the main problems faced by neurologists. A substantial amount of effort has been devoted to designing and manufacturing effective biomaterials to meet this challenge. Bioengineering and regenerative medicine both benefit from manipulating neuronal growth. Numerous factors affect the growth and function of neurons. Physical and chemical cues can influence neuronal growth. The spatial gradient of chemical repelling molecules and chemical attracting molecules has been revealed to influence neuronal growth processes towards their specific targets [131,132]. Nanoscale silver particle densities can also influence neuronal growth [133]. Nissan et al., recently studied the effect of the Ga@CDs@Ga NPs on the growth of the neurites (or neuron cell) during the initiation and elongation growth phases. It was found that cells were grown on a Ga@C-dots@Ga-coated substrate exhibited a 97% increment in the number of branches appearing from the soma. Also, surface modification and particle morphology play a significant role in neural growth [14]. Figure 11a explains the several coated glass surface and different morphological characteristics that were studied with SH-SY5Y cells. The lowest number of neurites was counted in the control sample (glass surfaces) without CDs and Ga@CDs (Figure 11b).

An increase of 40% in the number of neurites derived from the soma was observed when the cells were implanted on a Ga@C-dots@Ga-coated glass substrate (Figure 11c,d) [14]. The number of neurites in the other mixtures increased only by 10–20%. On top of that, it was observed that neuron cells grown on a Ga@C-dots@Ga NP-coated glass substrate exhibited a 94% increase in the number of branches emanating from the soma of neuron cells (Figure 11f). The HRSEM micrograph (Figure 11e,g) demonstrates that the neurites split and reach the NPs. The number of branches in CDs and CDs@Ga NPs increased by 40–50% [14]. When Ga was deposited on the CDs before the cell was deposited, the same number of splits in neurons was detected. Based on these results, the authors concluded that the size and the type of substrate on which the neuronal cell grows play a major role in neuron cell growth. It was also observed that when the neuron cells were grown on large pristine Ga spheres, this effect of neuron growth was not observed, but the doping of Ga in CDs slightly increased the growth of neuron/neurites. Thus, it is the nature of the materials and doping of Ga in CDs that influences neural cell growth.

### 4.7. Drug-Functionalized Hybrid CDs for Drug Delivery

Approximately 10 million people die from cancer every year, and 18 million new cases are reported, according to the WHO (World Health Organization [134,135] There has been a significant increase in the number of drug-resistant tumors and tumor relapses even as significant advances have been made in therapeutics and diagnostics [135]. The current treatments consist of chemotherapy, radiation, and surgery. The conventional chemotherapy treatments do not provide accurate results and carry serious risks. Their side effects can be severe, and they do not minimize metastasis [136]. In the event that single treatments fail, combination therapies are often administered, which can increase the hazard of serious toxicities. Furthermore, several carcinogenic molecules, such as hydrophobic drugs, radioisotopes, toxins and disordered nucleic acids, could not be injected systemically to the patients due to their instability or extensive off-target effects. Another issue with these systems is their lack of selectivity. In recent studies, 80% of patients were found to have a poor response to conventional therapy and become resistant to it, which ultimately increases their healthcare costs [137], Generally, conventional treatments cause fluctuations in plasma drug concentrations, which are harmful to health. Therefore, anti-tumor drugs need to be optimized for therapeutic efficacy with anticancer functionalized hybrid CDs [2,121,138], and cancer treatments need to be improved [135]. In recent studies, it has been demonstrated that intratumoral delivery of drugs through direct injection into a tumor mass can deliver extremely large amounts of drugs at the target site while minimizing systemic toxicity since intravenous chemotherapy is associated with severe complications and treatment abandonment [139,140] Several researchers have developed the drug functionalized or doped CDs-based drug delivery system for intralesional injection of CDs-DOX for the treatment of liver, kidney, lungs and other tissue cancer in mice. For example, Sun et al., have recently designed the functionalization of CDs with doxorubicin (DOX) and made the CDs-DOX composites for the treatment of cancer cell [141]. DOX was bind covalently to the surface of CDs particle for drug delivery (Figure 12a) [141]. Overall, cellular/tissue uptake and intracellular trafficking mechanisms of DOX-hollow CDs were clearly demonstrated (Figure 12b) by Wang et al., in 2013 [142]. There are possibilities to enter the DOX-CDs nanocomposites into the cell or tissue by endocytosis and formed vesicles (vesicles will transport into the lysosomes). Finally, in the acidic environment of the lysosomes, the protonated DOX can be released and then enter into the nuclei (Figure 12b) [142]. The CDs aqueous solution was injected subcutaneously into a nude mouse and imaged at 488 and 535 nm emission filter [142]. Figure 12c shows strong greenish fluorescence at the administered place (spot), suggesting that the DOX-CDs composites fluorescence penetrated the skin/tissue of the mice. Furthermore, the mice remained healthy after injection, indicating that the CDs-DOX nanocomposites had exceptional biocompatibility and low toxicity for the animals model [141]. Based on these results, Sun et al., used drug functionalized or doped CDs for bioimaging both in vitro and in vivo [141]. Additionally, the nanoprobe was injected into the tail vein to measure the biodistribution and excretion of the drug functionalized CDs. Fluorescence imaging of various organs was performed at various time points (0, 0.5, 1, 3 h) [141]. Figure 12d shows that the mice kidney and mice bladder demonstrated high fluorescence signals after injection than other organs like the heart, spleen and liver. The results indicated that drug-functionalized CDs could be eliminated by the kidney and bladder systems [143].

Sun et al., prepared DOX-entrapped CDs for the imaging and enhancement of inter/intracellular drug delivery. They showed that DOX-CD nanocomposites have well-developed crystal structure, outstanding aqueous stability, and excellent fluorescence behavior with a very good quantum yield of 93% [141]. They observed that DOX-CDs nanocomposites were easily absorbed by the cancer cells and label them so that they could by detected by fluorescent images [141]. Further, endo-lysosomal pH-mediated DOX release behavior was analyzed via the DOX-CDs system, and no cytotoxicity of the DOX-CDs was observed using the MTS assay against the H0-8910 ovarian cancer cell line [141]. Therefore, Sun et al., concluded that the DOX-CDs nanocomposites could be a promising probe for live cell imaging and intracellular drug delivery. Similarly, confocal fluorescence images were obtained for HepG2 cells and HL-7702 cells that were co-incubated with DOX-CDs for three hours in order to observe DOX release in vitro [143]. Figure 12e shows that the reddish fluorescence of DOX molecules is only observed in HepG2 cells but not in the HL-7702 cells. Based on this result, CD–DOX emits mainly in green, as opposed to red-fluorescence-free DOX. In cancerous HepG2 cells, the bound DOX molecules are released, unlike in HL-7702 cells. [143]. The same concept was applied to the bio-distribution of the CDs-DOX nanoformulation. The results indicated that both DOX and DOX-CDs nanoformulation can easily enter into the tumor cells to acts as anticancer/antitumor drugs. The photographs taken at various time intervals (Figure 12f) show the reduction in the tumor volume in a BALB/c-nude mouse treated with CDs–DOX nanoformulation. A comparison between the green-emitting CDs in the treated mouse to an untreated mouse demonstrates the efficiency of CDs as in-vivo fluorescent tracing agents to determine drug biodistribution. Figure 12g presents the fluorescence emission (excitation at 405 nm) from the tumor site of a mouse injected with functionalized CD–DOX formulations, in comparison to a non-injected mouse (both under 405 nm line excitation) [143]. Based on these results, the targeted therapeutic function of the CD-DOX nanoformulations could be universal to all malignant tumour cells.

## 5. Limitations of Doped CDs and Future Prospects

Doped or functionalized CDs have gained significant recognition as excellent fluorescence candidates compared to semiconductor quantum dots because of their unique and enviable physical-chemical properties. The origin of the fluorescence emission of metal/nonmetal-doped CDs is the subject of substantial debate, and further research is needed. Particularly, the optical and biological properties of doped CDs are affected by the synthesis route, as well as by the precursors used (doping amount of metal or nonmetal, starting carbon precursor, reaction temperature, and reaction time) due to their inherent morphology and structure. By using a top-down process, multi-morphology carbon structures with high crystallinity and relatively intact structures produce blue to red-colored doped CDs. In order to increase their luminescence efficiency, however, surfaces must usually be modified. Nevertheless, some issues such as the precise origin of fluorescence, the role of metal/nonmetal doping in CDs, and their location in the CDs are still unresolved. Additionally, the development of facile and effective approaches for preparing scalable larger amounts and high-quality metal/nonmetal or hybrid CDs for industrial production is still in its infancy. In addition, multi-color CDs have a low selectivity due to their unclear intrinsic structures and surface groups. Recently, advancements in the selection of appropriate carbon precursors and the development of facile methods have facilitated the search for an all-in-one strategy to produce multi-colored CDs. Accordingly, as doped CD-based materials chemistry continues to develop rapidly, a wider range of applications in related fields is expected in the near future.

## 6. Conclusions and Summary

Reviewing several applications of metal- and nonmetal-doped/hybrid CDs, particularly in biomedical fields, was the purpose of this article. The goal was to provide an overview of recent advances in metal and nonmetal doping of CDs, including synthesis approaches, physical-chemical properties, and biomedical applications. Furthermore, metal ion/nonmetal ion doping has been demonstrated to be a promising synthetic route to enhance the chemical, optical, electrical, and magnetic properties of doped CDs by modifying the electronic and atomic arrangement of the CD structure. Thus, doped CD-based nanomaterials have demonstrated remarkable performance in nano probing, sensing, imaging, drug delivery, and tissue engineering. Researchers have significantly improved the biomedical performance of nonmetal/metal-doped CDs by using innovative synthesis techniques. Due to their unique fluorescent properties, excellent biocompatibility, and high aqueous stability, doped C-dots can be a versatile material for biomedical and sensing applications.

## Figures and Tables

**Figure 1 nanomaterials-12-00898-f001:**
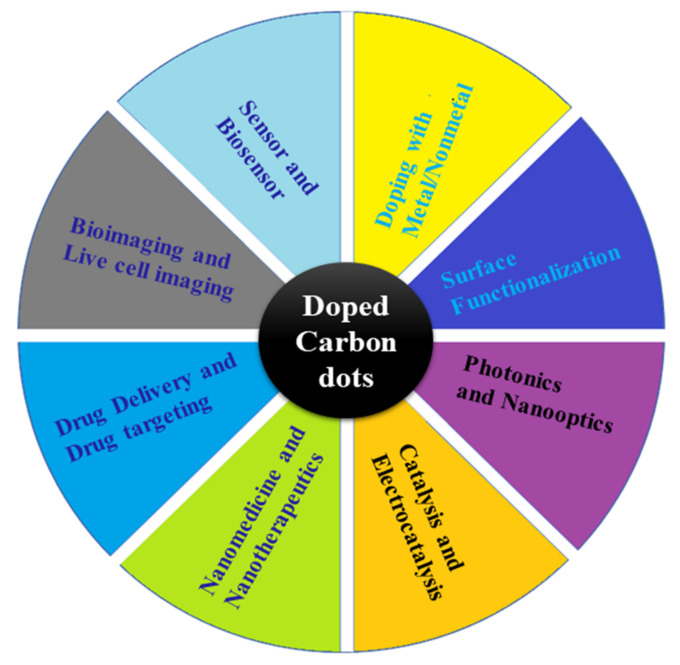
Overview of doped and hybrid CDs and showing the possible potential applications.

**Figure 2 nanomaterials-12-00898-f002:**
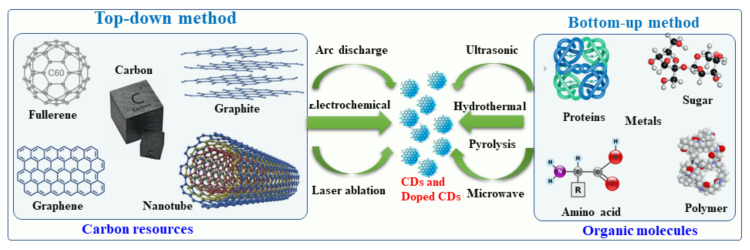
This diagram illustrates the top-down and bottom-up approaches to the synthesis of CDs.

**Figure 3 nanomaterials-12-00898-f003:**
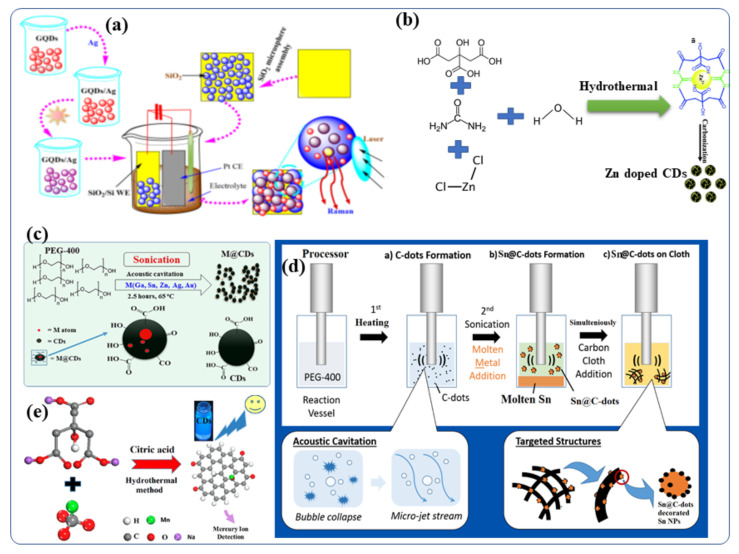
Schematic presentation of the synthetic procedures for Metal-doped CDs: (**a**) silver-doped CDs [5,64] (**b**) Synthesis of Zn-doped CDs, including the dehydration, coordination, and carboncarbonation steps. It also illustrates applications of the Zn doped CDs in fluorescent hybrid materials for inkjet printing [69]. (**c**) Synthesis of CDs and metal-doped CDs protocol [12]. (**d**) Sonochemical synthesis route of Sn@C-dots and Sn@C-dots@Sn nanoparticles [80]. (**e**) Synthetic process for the Mn-doped CDs [76]. Reproduced with copyright permission.

**Figure 4 nanomaterials-12-00898-f004:**
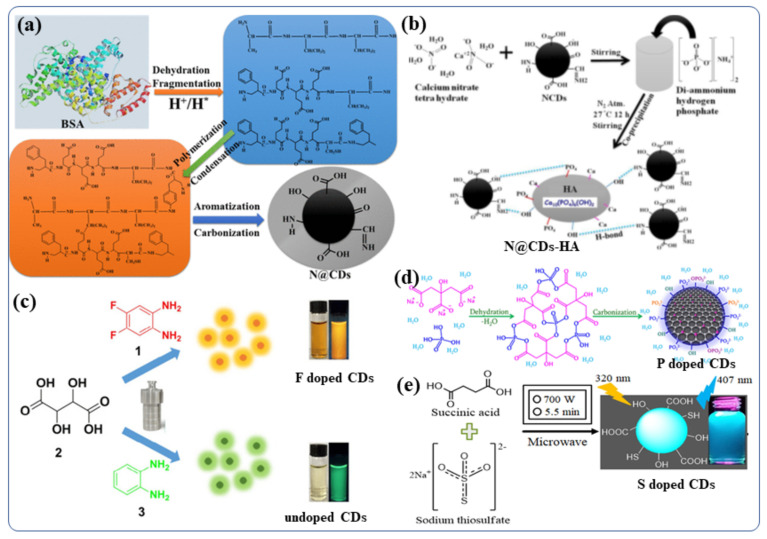
Synthetic procedures for non-metal doped CDs (**a**) NCDs (N@CDs) [88]. (**b**) NCDs-HA nanoparticles and their possible binding mechanism with HA particles [89]. (**c**) F-CDs and undoped CDs [94]. (**d**) P-doped CDs [97]. (**e**) S-CDs nanomaterials [36]. Reproduced with copyright permission.

**Figure 5 nanomaterials-12-00898-f005:**
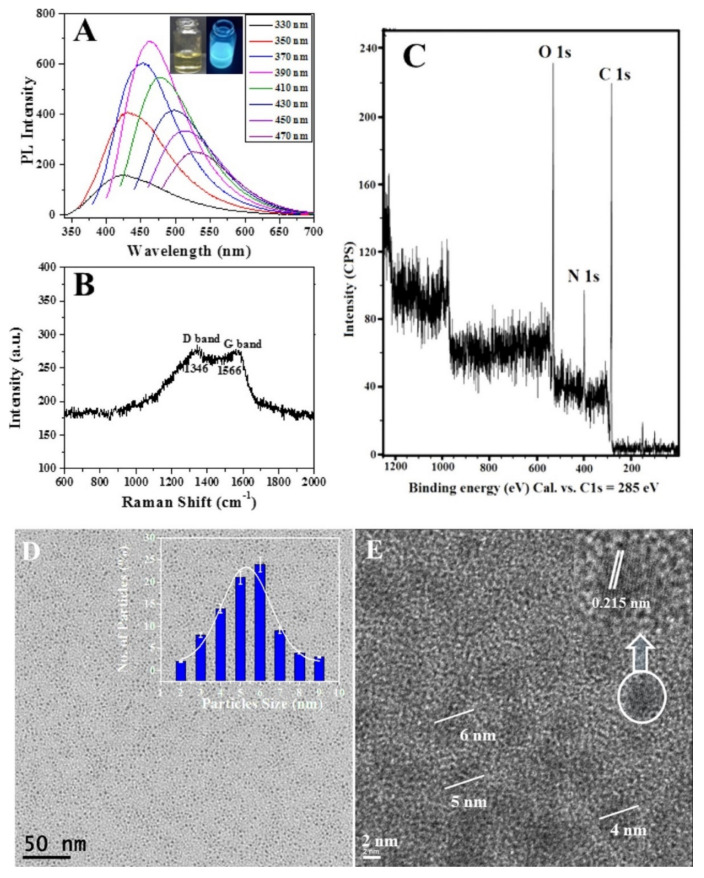
(**A**) The fluorescence spectrum of N-doped CDs dissolved in water. (**B**) Raman spectrum of the powder of N-doped CDs on a glass slide. (**C**) Full XPS spectrum of N-doped CDs. (**D**) TEM images of a cluster of N-doped CDs. Inset: Particle size-distribution. (**E**) HRTEM image of the N-doped CDs. Inset: the crystal lattice of a single N-doped CDs particle. Copyright with permission from Ref [88]. (2017), from American Chemical Society.

**Figure 6 nanomaterials-12-00898-f006:**
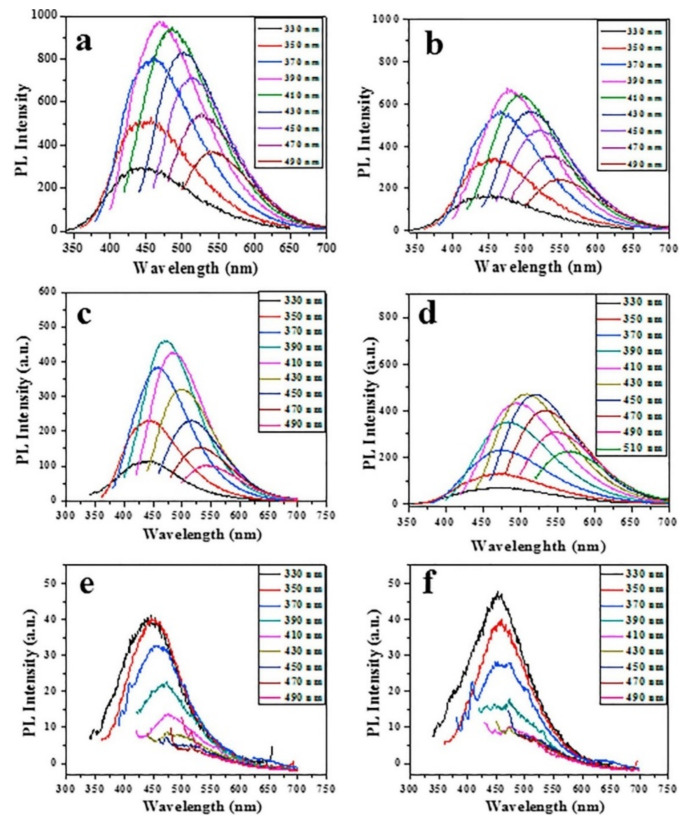
Fluorescence spectroscopy of (**a**) Pristine CDs (**b**) Ga-doped CDs, (**c**) Zn-doped CDs, (**d**) Sn-doped CDs, (**e**) Ag-doped CDs, and (**f**) Au-doped CDs at various excitation wavelengths. Copyright with permission from Ref [12]. (2019), from Elsevier.

**Figure 9 nanomaterials-12-00898-f009:**
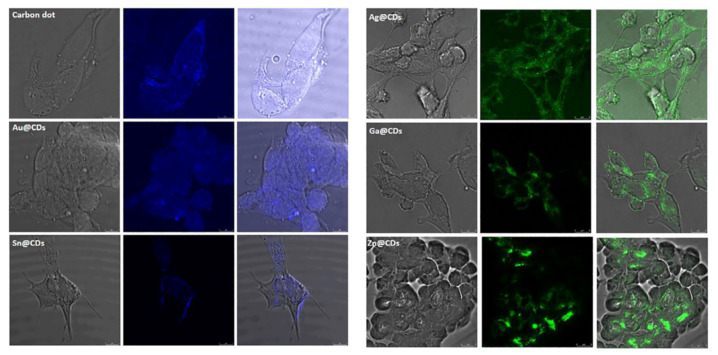
Confocal microscope images of neuroblastoma cells (neuronal cell, SH-SY5Y) with metal-doped (Ag, Ga, Au, Sn, and Zn) CD-formulated samples. Scale bar = 10 μm. Reproduced from ref. [12], copyright (2019), with permission from Elsevier.

**Figure 10 nanomaterials-12-00898-f010:**
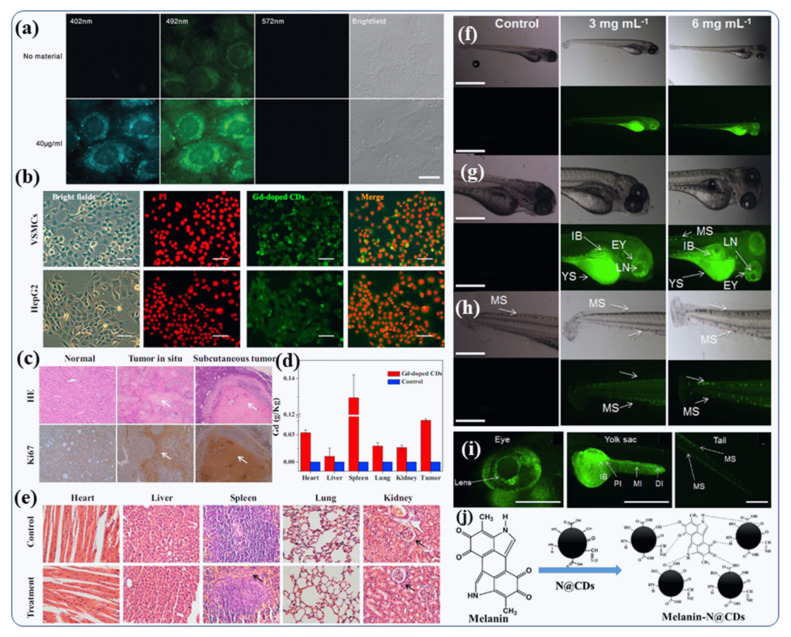
(**a**) Mitochondrial examination using N@CDs [15]. (**b**) Confocal microscopy images of PI and Gd-doped CDs stained on 4T1 and A549 cells (Bars are 50 m) [70]. (**c**) Methods for detecting subcutaneous tumors and tumors in situ by histopathology and immunohistochemistry [70]. (**d**) The H & E micrograph of major organs after giving the intravenous injection of Gd-CDs into the tail [70]. (**e**) Distribution of Gd ion/metal in the organs of mice bearing Herps tumors at the 24 h following intravenous injection [70]. (**f**) Zebrafish larvae (78 hpf) exposed to N@CD solutions after soaking in bright-field image (upper part) and fluorescence images (lower part) [88]. Upon enlarging the image, one can see (**g**) the lens, eye, tail, intestine, yolk sac, and (**h**) the part of melanophore strips [88]. There is a scale bar of 1.6 mm. (**i**) IB: intestinal bulb, YS: yolk sac, MS: melanophore strip, EY: eye, and LN: lens. (**j**) Possible mechanisms by which N@CDs bind to melanin. Reproduced with copyright permission.

**Figure 11 nanomaterials-12-00898-f011:**
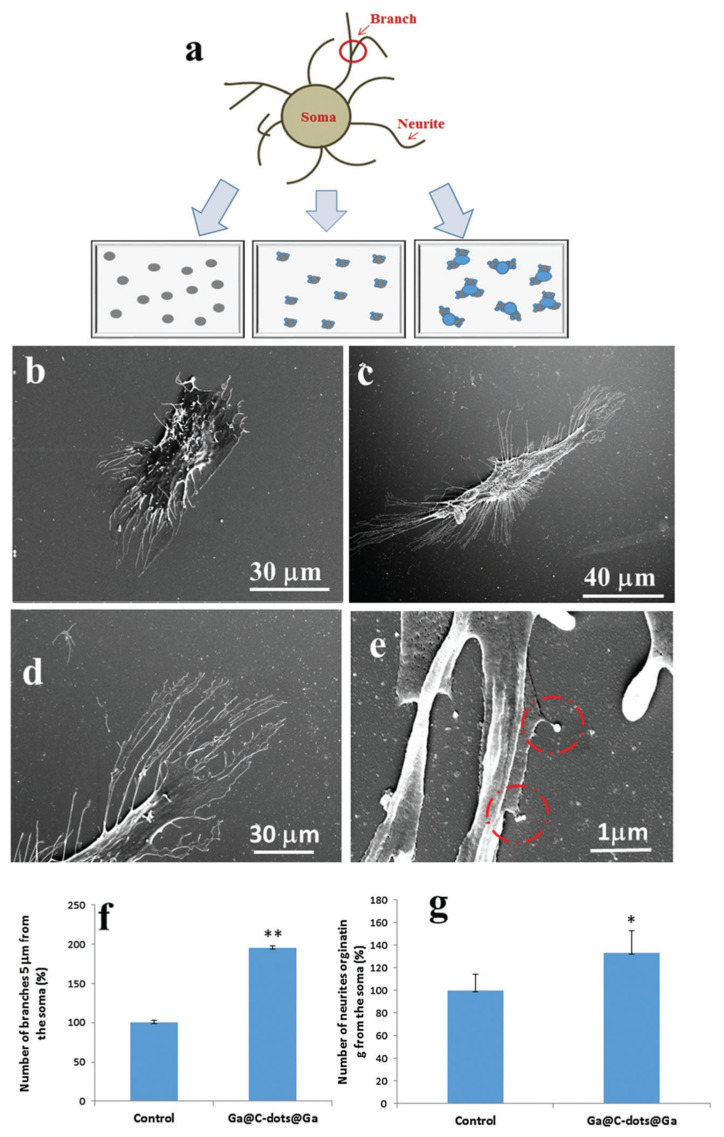
Morphology of cells. (**a**) A picture of SH-SY5Y cells showing neurites and branches on a modified glass substrate. (**b**,**c**) An HRSEM image of SH-SY5Y neuron cells on glass and a Ga@C-dots@Ga glass substrate. (**d**,**e**) Images obtained by HRSEM of neurites and their interaction with Ga@C-dots@Ga NPs. (**f**) There are SH-SY5Y branches located 5 mm from the soma. (**g**) The soma sends a number of neurites. Every experiment was performed 3–4 times (* *p* 0.05, ** *p* 0.005, *t*-test). Reproduced with copyright permission [14] from Royal Society of Chemistry.

**Figure 12 nanomaterials-12-00898-f012:**
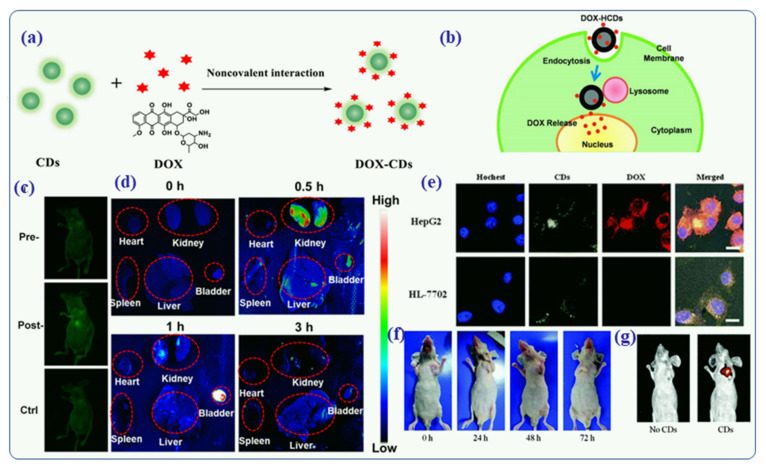
(**a**) Schematic representation of functionalized DOX-CD covalent ineteractions [141]. (**b**) The DOX-hollow CDs drug delivery system illustrates the intracellular release of DOX [142]. (**c**) An animal fluorescent imaging test using CDs was performed in vivo [141]. In (**d**), mice were imaged ex vivo after intravenously injected CDs at various times [141]. (**e**) under confocal microscopy, CD-DOX is endocytosis and then released into the cytoplasm. During cell nucleus death, Hoechst is responsible for the blue fluorescence, while CDs produce green-yellow fluorescence, and DOX drugs themselves are responsible for the red fluorescence [143]. (**f**) After intralesional injection, a photo of the HCC tumor site can be seen for a different period of time [143]. (**g**) Image of an in vivo fluorescence imaging of C-nude BALB/c-nude mice bearing HCC [143]. Reproduced with copyright permission.

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
