# Peer review of "Synthesis of Doped/Hybrid Carbon Dots and Their Biomedical Application"

_nanomaterials, 2022, doi:10.3390/nano12060898_

Round 1

Reviewer 1 Report

This work reviewed the CDs for biomedical applications. I think this work can be accepted after a major revision. 

  1. Define acronyms when they first appear; thereafter directly use them.
  2. The authors spend quite a lot of space on the synthesis and properties of CDs, and I think these should be reflected in the title. Therefore, I suggest the authors to make some changes to the title.
  3. In the introduction of sensor-related work, many works are actually not he biomedically relevant, such as heavy metal ion detection. These works fall under the scope of environmental sensor. I think the authors should have chosen the literature they used more carefully.
  4. The perspectives section is too short. I think you may added some summary with the perspectives.

Author Response

Reviewer 1

Comment 1. English language and style
( ) Extensive editing of English language and style required ( ) Moderate English changes required (x) English language and style are fine/minor spell check required ( ) I don't feel qualified to judge about the English language and style

Response: Thank you for your constructive comments. All grammatical and stylistic errors have been corrected.

Comments and Suggestions for Authors
This work reviewed the CDs for biomedical applications. I think this work can be accepted after a major revision.

Comment 1. Define acronyms when they first appear; thereafter directly use them.

Response: We have defined the acronyms in advance of their use in the manuscript.

Comment 2. The authors spend quite a lot of space on the synthesis and properties of CDs, and I think these should be reflected in the title. Therefore, I suggest the authors to make some changes to the title.

Response: The title of the paper has been revised accordingly.

Comment 3. In the introduction of sensor-related work, many works are actually not he biomedically relevant, such as heavy metal ion detection. These works fall under the scope of environmental sensor. I think the authors should have chosen the literature they used more carefully.

Response: An appropriate modification has been made to the introduction.

Comment 4. The perspectives section is too short. I think you may add some summary with the perspectives.

Response: The limitations and future perspectives section has been updated with more constructive information, as well as a summary and conclusion section.

Reviewer 2 Report

In this work, Kumar and co-workers reviewed advances in doped and hybrid CDs and their application in the biomedical field. This is a topic well within the scope of this journal, and should be of interest for both specialists and non-specialists. Furthermore, the authors performed a relatively good job in addressing current knowledge in synthesis, characterization and associated biomedical applications of metal/non-metal doped/hybrid CDs.

Nevertheless, there are aspects that should be corrected in Major Revision:

-Section 5, titled "Limitations and future perspectives", does not provide any real limitations and future perspectives/directions of the field. It should be totally reformulated to correct this;

-The authors must expand in the differences of metal and non-metal doping, and how and why they affect the properties of CDs. In the current version of the manuscript, these strategies are almost treated as if they are the same/lead to similar outcomes;

-CDs are generally more attractive than metal-based nanoparticles due to sustainability and safety features, which arise from their non-metal nature. Does metal-doping not negate these advantages? Furthermore, what are the advantages of metal-doping that cannot be obtained by the use of metal-based nanoparticles from the start? This must discussed in more detail.

-In the first paragraph of Section 4, the authors refer some potential of CDs in optoelectronics but do not expand on that on subsequent sub-sections;

-Lines 203-204: there is an author note that must be removed;

-Lines 61-63: bottom-up methods are referred, but microwave-assisted synthesis is not mentioned. This should be corrected, and citations added (as examples, DOIs: 10.1016/j.jclepro.2020.120080; 10.1021/acsami.8b13217).

-Lines 63-65: authors state that top-down methods are preferred (debatable) but only provide negative features of those processes.

-The quality of written English should be improved.

Author Response

Reviewer 2

Comment. English language and style

( ) Extensive editing of English language and style required (x) Moderate English changes required ( ) English language and style are fine/minor spell check required ( ) I don't feel qualified to judge about the English language and style

Response: We have corrected the English language and style, as you suggested.

Comments and Suggestions for Authors

In this work, Kumar and co-workers reviewed advances in doped and hybrid CDs and their application in the biomedical field. This is a topic well within the scope of this journal, and should be of interest for both specialists and non-specialists. Furthermore, the authors performed a relatively good job in addressing current knowledge in synthesis, characterization and associated biomedical applications of metal/non-metal doped/hybrid CDs.Nevertheless, there are aspects that should be corrected in Major Revision:

Comment 1. -Section 5, titled "Limitations and future perspectives", does not provide any real limitations and future perspectives/directions of the field. It should be totally reformulated to correct this;

Response: The limitations and future perspectives section has been updated with more constructive information, along with a conclusion and summary.

Comment 2. -The authors must expand in the differences of metal and non-metal doping, and how and why they affect the properties of CDs. In the current version of the manuscript, these strategies are almost treated as if they are the same/lead to similar outcomes;

Response: The difference between the synthesis of metal and non-metal doped carbon dots has been separately described (please refer to lines 127-236).

Comment 3. -CDs are generally more attractive than metal-based nanoparticles due to sustainability and safety features, which arise from their non-metal nature. Does metal-doping not negate these advantages? Furthermore, what are the advantages of metal-doping that cannot be obtained by the use of metal-based nanoparticles from the start? This must discuss in more detail.

Response: The paragraph has been rewritten to express the explanation in a more concise manner.

Comment 4. -In the first paragraph of Section 4, the authors refer some potential of CDs in optoelectronics but do not expand on that on subsequent sub-sections;

Response: We have rewritten and modified the first paragraph of section 4.

Comment 5. -Lines 203-204: there is an author note that must be removed;

Response: We have removed the suggested line.

Comment 6 -Lines 61-63: bottom-up methods are referred, but microwave-assisted synthesis is not mentioned. This should be corrected, and citations added (as examples, DOIs: 10.1016/j.jclepro.2020.120080; 10.1021/acsami.8b13217).

Response: We have Corrected and modified the sentences.

Comment 7. -Lines 63-65: authors state that top-down methods are preferred (debatable) but only provide negative features of those processes.

Response: We have Corrected and modified the sentences.

Comment 8. -The quality of written English should be improved.

Response: The English language and stylish has been improved.

Round 2

Reviewer 1 Report

The revised version can be accepted.

Reviewer 2 Report

The authors addressed my comments, and so, my recommendation is for acceptance. I would only point out that there are some sentences in red in the manuscript that I think to be comments made by the authors (like in page 10), which should be removed before publication.